# Chemistry of Peptide-Oligonucleotide Conjugates: A Review

**DOI:** 10.3390/molecules26175420

**Published:** 2021-09-06

**Authors:** Kristina Klabenkova, Alesya Fokina, Dmitry Stetsenko

**Affiliations:** 1Faculty of Physics, Novosibirsk State University, 630090 Novosibirsk, Russia; k.klabenkova@g.nsu.ru (K.K.); d.stetsenko@nsu.ru (D.S.); 2Institute of Cytology and Genetics, Russian Academy of Sciences, Siberian Branch, 630090 Novosibirsk, Russia

**Keywords:** cell-penetrating peptide, nucleic acid therapeutic, antisense oligonucleotide, small interfering RNA (siRNA), peptide nucleic acid (PNA), locked nucleic acid (LNA), phosphordiamidate morpholino oligomer (PMO), cellular uptake, drug delivery, click chemistry

## Abstract

Peptide-oligonucleotide conjugates (POCs) represent one of the increasingly successful albeit costly approaches to increasing the cellular uptake, tissue delivery, bioavailability, and, thus, overall efficiency of therapeutic nucleic acids, such as, antisense oligonucleotides and small interfering RNAs. This review puts the subject of chemical synthesis of POCs into the wider context of therapeutic oligonucleotides and the problem of nucleic acid drug delivery, cell-penetrating peptide structural types, the mechanisms of their intracellular transport, and the ways of application, which include the formation of non-covalent complexes with oligonucleotides (peptide additives) or covalent conjugation. The main strategies for the synthesis of POCs are viewed in detail, which are conceptually divided into (a) the stepwise solid-phase synthesis approach and (b) post-synthetic conjugation either in solution or on the solid phase, especially by means of various click chemistries. The relative advantages and disadvantages of both strategies are discussed and compared.

## 1. Introduction

The peptide-oligonucleotide conjugate (POC) is a name usually applied to a synthetic molecule constituting one or more residues of a linear or, less often, a cyclic peptide linked by a covalent bond to an oligonucleotide or its analog. As chimeric compounds that include an (oligo)peptide part and a nucleic acid part, each peptide-oligonucleotide conjugate (POC) represents a combination of its parent biomolecules, such as the immanent base-pairing ability of nucleic acids and the multifaceted bioactivity of the structurally and functionally diverse peptides. Although the compounds related to POCs occur in nature as nucleopeptides [1,2,3], this review, as it is focused on the chemical methods of conjugating peptides to oligonucleotides, will be necessarily limited to synthetic substances only.

The interest in peptide-oligonucleotide conjugates was sparked by the advent of antisense technology [4], followed by the development of the first generation of therapeutic oligonucleotides at the end of the 1980s [5,6]. After a period of research, it was generally accepted that a successful nucleic acid drug ought to demonstrate better cellular uptake than what the majority of the explored to-date oligonucleotide chemistries can offer [7,8]. This understanding coincided with the serendipitous discovery of what was later to be called cell-penetrating peptides in the mid-1990s [9].

Clinical application of therapeutic oligonucleotides officially started in 1998, when the US Food and Drug Administration (FDA) approved the first nucleic acid drug fomivirsen (Vitravene^®^) [10] for the treatment of cytomegalovirus-induced blinding retinitis in AIDS patients [11]. After the seminal work on RNA interference (RNAi) [12], it took over 20 years for the first small interfering RNA (siRNA) therapeutic patisiran (Onpattro^®^) to appear [13]. To date, the progress in non-clinical and clinical studies with synthetic oligonucleotides has prompted the FDA approval of a total of 12 drugs, whereas over 130 are going through various phases of clinical trials [14,15]. However, despite their huge therapeutic potential, oligonucleotides and their analogs, due to their intrinsic physicochemical characteristics, in most cases face the problem of ineffective transport through the cellular membrane, usually via an endocytotic pathway [16,17]. Another related although more specific problem is the delivery through the blood-brain barrier in the case of oligonucleotide therapeutics for neurodegenerative diseases [18,19]. Moreover, in addition to passage through the outer cellular membrane, it is necessary for oligonucleotides to escape from endosomes [20] and translocate to appropriate compartments, such as the nucleus [21].

One of the ways to overcome the limitations of poor cellular uptake is through the conjugation (a covalent attachment) of an oligonucleotide or its analogue to a moiety that promotes cellular penetration. A number of such carriers from small molecules to macromolecules and supramolecular assemblies have been proposed, such as cholesterol [22,23,24] and other lipids [25,26,27]; polymers [28], in particular, polyethyleneimine [29]; dendrimers [30]; inorganic nanoparticles [31]; DNA nanostructures [32,33]; and others [34]. However, despite the tremendous progress in non-viral nucleic acid delivery over the past 25 years [35], there is still no consistent solution for conjugation with such moieties that would be applicable to most cell types and a wide range of biological targets that can significantly improve the in vivo efficacy of the prospective oligonucleotide therapeutics.

Cell-penetrating peptides (CPPs) were identified over 20 years ago as one of the promising carriers for oligonucleotide delivery [36,37]. Due to their ability to penetrate into cells and mediate the delivery of such cargo molecules as non-cell-penetrating peptides [38], proteins [39,40], nanoparticles [41], quantum dots [42], and nucleic acids [43,44], CPPs gradually became an invaluable tool to increase the concentration of difficult-to-deliver macromolecules in certain cells, cell compartments, tissues, and organs [45]. The CPP could be employed either as a non-covalent additive, which may self-assemble into peptide nanoparticles to encapsulate cargo [46,47], or as a covalently attached moiety in the form of a peptide conjugate [48,49]. The peptide-oligonucleotide conjugates (POCs), which will be covered in this review, were employed as vehicles for oligonucleotide delivery in various medicinal applications, such as antimicrobial, antiviral, anticancer, or splice-switching therapies [50,51]. Recent studies have shown that conjugates of CPPs with various oligonucleotides and, more often, their analogues demonstrate excellent efficiency, in particular, as antibacterials or as splice-switching agents for such genetic diseases as Duchenne muscular dystrophy or spinal muscular atrophy [52,53]. Thus, an overview of the currently applicable methods for the chemical synthesis of peptide-oligonucleotide conjugates, with particular emphasis on more recent developments, would be useful for an in-depth understanding of this highly promising area of oligonucleotide therapeutics research.

## 2. Nucleic Acid Therapeutics

Currently, nucleic acid derivatives are considered powerful tools for treating various diseases at the posttranscriptional level. Contrary to small-molecule drugs, oligonucleotides, which are short, synthetic single- or double-stranded DNA, RNA, or their analog sequences, have the unique ability to recognize and bind in a selective way to the complementary sequences of (predominantly) cellular RNAs, including pre-mRNAs, mRNAs, and noncoding RNAs [54], such as micro-RNAs [55] as well as viral or microbial RNAs [56,57]. Moreover, genomic DNA, proteins, and even small biomolecules could be targeted by oligonucleotide derivatives, such as triple-helix-forming probes, DNA decoys, and nucleic acid aptamers [58,59,60,61]. Thus, nucleic acid therapeutics may affect biological processes in which target genes and their expression products are involved, interfere with pathogen metabolism, modulate the immune response to certain antigens, etc. There is a wide variety of therapeutic oligonucleotide classes, such as antisense oligonucleotides, small interfering RNAs (siRNAs) [62], ribozymes [63], deoxyribozymes (DNAzymes) [64,65], antagomirs [66], and guide RNAs for CRISPR/Cas9 [67], that, despite their great variety, have a common feature in the mechanisms of their action, which is complementary base pairing [68].

### 2.1. Antisense Oligonucleotides (ASOs)

Historically, antisense oligonucleotides (ASOs) were the earliest and, currently, the best-studied class of nucleic acid therapeutics. The concept of ASOs originated in 1978, when Zamecnik and Stephenson demonstrated that a specific 13-mer oligodeoxynucleotide inhibited Rous sarcoma virus replication in chicken embryos [4]. The mechanism of the therapeutic effect of ASOs rests on the ability of synthetic oligonucleotides or their analogues to bind to a complementary RNA through the canonical Watson–Crick duplex to alter the metabolism of the corresponding RNA in one of the following ways (Figure 1).

A more general way for ASOs to interfere with RNA function, e.g., the initiation or elongation of translation of an mRNA, is to physically shield a specific fragment of a regulatory region of the RNA, e.g., the translation initiation site, by forming a duplex with ASOs (steric block) [69,70,71,72]. This approach is particularly applicable when one needs to preserve the functional RNA, e.g., in the case of splicing redirection of a pre-mRNA by a splice-switching oligonucleotide [73,74,75]. Another way is to activate enzymatic RNA digestion by recruiting a cellular RNase, most commonly RNase H [76], to hydrolyze the RNA strand of the ASO-RNA duplex [77].

The first ASOs to be investigated were native oligodeoxynucleotides (Figure 2, **1a**) that proved to be rapidly digested by nucleases in the serum unless protected by at least minimal chemical modification [78,79]. Thus, unmodified oligonucleotides proved to be unsuitable for in vivo applications. For this reason, a range of chemical modifications were introduced into ASOs to render the prospective oligonucleotide therapeutics sufficiently resistant to enzymatic hydrolysis of the internucleotidic phosphodiester bond (Figure 2) [80]. Therefore, the first-generation ASOs may be said to incorporate the modified phosphate linkages, such as phosphorothioate (**1b**) [81], methyl phosphonate (**1c**) [82], more rarely phosphorodithioate (**1d**) [83] and boranophosphate (**1e**) [84], and recently reported mesyl phosphoramidate (**1g**) [85,86], as well as many others [87,88]. Another group of ASOs consists of oligonucleotides with modifications in the ribose ring that not only offer a varying degree of protection against nucleases but, even more importantly, increase the stability of the ASO-RNA duplex [89,90,91], notably 2′-*O*-methyl (**2b**) [92,93,94], 2′-*O*-(2-methoxy)ethyl (MOE) (**2c**) [95,96], 2′-deoxy-2′-α-fluoro (**4**) [97], and, especially, constrained ribose analogues such as bridged/locked nucleic acids (B/LNAs) (**3**) [98,99,100,101] and tricyclo-DNAs (**5**) [102]. A separate class of ASOs encompasses oligonucleotide analogs, in which the natural ribose-phosphate backbone is replaced by a suitable surrogate; typical examples would be peptide nucleic acids (PNAs) (**6**) [103] and phosphordiamidate morpholino oligomers (PMOs) (**7**) [104,105]. The latter, in particular, gave rise to the three splice-switching oligonucleotide drugs for the treatment of Duchenne muscular dystrophy approved by the FDA in 2016-2021: eteplirsen (Exondys 51^®^) [106], golodirsen (Vyondys 53^®^) [107], and casimersen (Amondys 45^®^) [108].

### 2.2. Small Interfering RNAs (siRNAs)

Small interfering RNAs (siRNAs) are (usually) double-stranded oligoribonucleotides (as in Figure 2, **2a**) with a length of 20–25 nt per strand, which were found in plants in 1999 [109]. The year before, Fire and Mello discovered a natural process of specific gene silencing termed “RNA interference” (RNAi) that was mediated by short double-stranded RNAs (including siRNAs) via a mechanism that is notably different from the antisense mechanism (the Nobel Prize in Physiology and Medicine of 2006) [12]. Later, Tuschl and coworkers demonstrated that synthetic siRNAs are able to induce RNAi in mammals [110].

A typical siRNA has dinucleotide overhangs at the 3′-end of each strand. One strand that is complementary to a specific region of the target mRNA is usually called the antisense strand, while the other one is called the sense or passenger strand [111]. In nature, this structure results from the action of the Dicer enzyme, which cleaves long double-stranded RNAs or short hairpin RNAs into siRNA duplexes (Figure 3) [112]. Then, in the RNA-induced silencing complex (RISC) with the participation of the Argonaut protein Ago2, the siRNA duplex is unwound, and the complementary duplex of the antisense strand with the concomitant mRNA is formed, followed by degradation of the latter. This results in potent expression downregulation for the corresponding gene via translation arrest at the mRNA level, similarly to that of the antisense mechanism (Figure 3).

As the origin and progression of many diseases are associated with upregulation of a particular gene, the use of synthetic siRNAs for therapeutic gene silencing is of great interest [113]. However, siRNA delivery to specific tissues, with the notable exception of the liver via the respective GalNac conjugates [114], remains an obstacle on the way to the clinics. Nevertheless, the recent FDA approval of two more therapeutic siRNAs (apart from the pioneering patisiran), givosiran (Givlaari^®^) [115] and lumasiran (Oxlumo^®^) [116], as well as one more approved by the European Medicines Agency (EMA), inclisiran (Leqvio^®^) [117], is indicative of the great promise offered by this particular area of drug development.

### 2.3. CRISPR/Cas9

The clustered regularly interspaced short palindromic repeats (CRISPR) were first discovered in *E. coli* in 1987 [118], but their detailed study only began in 1993 by Francisco Mojica [119]. Later, Jansen et al. investigated that near the CRISPR locus, there is always a set of homologous genes called CRISPR-associated systems or Cas genes that encode endo- or exonucleases [120]. Although CRISPR/Cas systems were found in a large number of prokaryotes, almost nothing was known about their function until 2005, when Mojica et al. published a paper showing the relationship of CRISPR loci with adaptive immunity in prokaryotes [121]. Several further studies have shown that between repeats in loci, there are different DNA “spacers” corresponding to parts of the viral genomes corresponding to past parasites of these bacteria [122]. Thus, spacers carry inherited memories of past cellular invasions. CRISPR RNA (crRNA) is transcribed from these spacers and directs Cas proteins to the foreign viruses, causing the cleavage of the foreign DNA [123]. In addition, it has been shown that Cas proteins need a special sequence localized near the target DNA, called a protospacer adjacent motif (PAM), for recognition and binding to the target [124].

From all the variety of CRISPR/Cas systems, scientists were most interested in the type II system from *Streptococcus pyogenes* for therapeutic application in genetic engineering, since only one Cas9 protein is required for its full operation [125]. In addition to Cas9, this system requires the presence of crRNA and trans-activating CRISPR RNA (tracrRNA) [126], which together form a duplex that directs Cas9 endonuclease to the target. Later, Doudna and Charpentier with colleagues designed a system that included only two elements, Cas9 and chimeric RNA combined from two molecules crRNA and tracrRNA, called a single-guide RNA (sgRNA) [127]. With such a system, it became possible to direct Cas9 to any DNA sequence for its cleavage only by changing the nucleotide sequence of sgDNA. The work was deemed so significant that it was awarded a Nobel Prize in Chemistry in 2020. The possibility of using the CRISPR/Cas9 system in eukaryotic cells has been demonstrated [128,129,130]. It was also shown that in eukaryotic cells, after CRISPR/Cas9-mediated double-stranded DNA breaks, the DNA molecule is not degraded, but rather repaired by two main pathways, namely non-homologous end-joining (NHEJ) and homology-directed repair (HDR) [131]. HDR is preferred because it allows the desired nucleotide sequence to be obtained by using an exogenous template as a recombination donor. Currently, many variants of the Cas9 protein have been developed [132,133,134].

Today, in most cases, a clinical application of CRISPR is based on ex vivo gene editing of cells with their subsequent re-introduction into the patient [132]. The ex vivo editing approach is highly effective for many diseases, including cancer and sickle cell disease. In turn, in vivo editing is largely limited by the lack of availability of the target tissue or organ. Despite this, recently a CRISPR-modified virus was injected into the patient’s eye in an attempt to treat Leber congenital amaurosis [133]. However, before widespread application of CRISPR technology in clinical practice, it is necessary to carry out many more experiments to make final conclusions on the effectiveness and safety of this method in vivo.

### 2.4. The Problem of Oligonucleotide Delivery

In contradistinction to small-molecule drugs, oligonucleotides are macromolecules, and their physicochemical properties, in particular, their polarity and polyanionic nature of the ribose phosphate backbone, essentially prevent passive diffusion through the phospholipid bilayer of a biological membrane. Thus, overcoming a problem of selective delivery of a nucleic acid drug to the right organ/tissue after systemic or local administration, followed by efficient transport into the specific cells and, once inside the cell, translocation to the correct cellular compartment to find its molecular target, is a keystone of oligonucleotide-based therapy. On the way to bind a unique RNA, the oligonucleotide ought to cross a number of extracellular and intracellular barriers, which have been extensively reviewed by Juliano and coauthors [134,135,136,137] and others [138].

It is believed that oligonucleotides are taken up into cells via receptor-mediated endocytosis [139]. Therefore, there is a need for an oligonucleotide therapeutic to escape from endosomes into the cytosol to trigger RNAi (for siRNAs), or reach the nucleus for splice-switching and RNase H activation [140,141]. All the way from the initial administration to the ultimate site of therapeutic activity, the oligonucleotide may be attacked by various exo- and endonucleases [142,143,144]. These are the main obstacles on the way to the successful clinical application of therapeutic oligonucleotides.

Thereby, it becomes an important task to design special delivery vectors for the effective transport of nucleic acid drugs into the cytosol and nucleus. Viral, e.g., adenoviral, vectors have been developed as specific carriers for nucleic acids for gene transfer and gene therapy [145]. However, despite several approved to-date gene therapies [146,147], there are still considerable limitations due to immunogenicity and safety concerns. Mainly, the application of a viral vector to deliver cargo to human cells induces an immune response. Thus, repeated administration of the same viral constructs becomes useless [148].

Thus, non-viral vectors have received widespread attention as an alternative delivery strategy that could ensure safe, efficient, and addressable oligonucleotide delivery. The non-viral methods traditionally include the use of liposomes [149], polymers, dendrimers [150], inorganic nanoparticles, or conjugation to certain small molecules [151]. Among the above, cell-penetrating peptides have become one of the most promising carriers to help oligonucleotides to translocate through cellular barriers via either covalent (peptide conjugate) or non-covalent (peptide additive) association.

## 3. Peptide-Mediated Cellular Delivery: A Brief Overview

The term “cell-penetrating peptide” (CPP) was introduced by Langel and coauthors [152] and usually refers to a short- to medium-size peptide containing between 5 and 40 amino acids. A CPP can pass through cell membranes through energy-dependent or energy-independent mechanisms, and moreover, it can facilitate the intracellular transport of various cargo molecules, which are poorly able to cross the membranes alone, such as other (non-cell-penetrating) peptides, proteins, nanoparticles, or nucleic acids [153].

The first CPP was discovered over 30 years ago at the end of the 1980s. Two research groups, when studying the activity of the transactivation transcription activator (Tat) domain of HIV-1, independently noticed that it can be efficiently internalized by cells in vitro [154,155]. A few years later, the Proschiantz group, when studying the role of *Drosophila* homeodomain proteins in post-mitotic neurons, discovered that a 60-amino-acid homeodomain protein sequence of the Antennapedia gene was able to cross biological membranes by an energy-independent pathway. The discovery led to the study of the ability of a series of synthetic peptides derived from the third helix of the Antennapedia homeodomain to be internalized by cells. In particular, it was shown that a 16-mer peptide named penetratin (pAntp) successfully translocated into cells, while shorter peptides were not internalized [156].

Later, Lebleu and coauthors probed the sequence of Tat protein to ascertain which sequence may be responsible for its cellular uptake. To achieve this, several peptides from residues 37–60 of the Tat domain were synthesized. As a result, a shorter version of Tat peptide 13 amino acids in length, located from amino acids 48 to 60, was identified as necessary for penetration into cells [157].

In 1998, the successful application of pAntp for in vivo delivery into Bowes cells of 21-mer PNA blocking the expression of the galanin receptor was demonstrated [158]. One year later, the Tat peptide was used for in vivo delivery of β-galactosidase [159]. These studies demonstrated the potential of CPPs for the in vivo delivery of cargo macromolecules, which is being extensively studied up to now to transport oligonucleotides, their analogs, and other difficult-to-deliver potential therapeutics across cellular membranes [160,161].

## 4. Cell-Penetrating Peptides (CPPs): Types and Examples

At different times, various criteria based on the sequence, function, or penetration mechanism have been proposed for classification of CPPs. However, there is currently no single taxonomy of these peptides. There are two CPP classifications in the literature: one that is based on the origin of peptides and the other one based on their physicochemical properties.

By their origin, the peptides are classified into protein-derived ones, such as Tat or penetratin; synthetic, such as polyarginine R8; and chimeric, which are combined from peptide fragments with different properties, such as transportan. This type of classification is not quite convenient and is mostly historical because it does not allow one to evaluate CPPs from the point of view of their interaction with cells.

According to their physicochemical properties, CPPs are broadly divided into three main classes: cationic, amphipathic, and hydrophobic peptides.

### 4.1. Polycationic CPPs

Polycationic peptides, as the name suggests, consist predominantly of positively charged amino acid residues, such as Arg, Lys, His, or, more rarely, Orn and others. This polycationic nature of peptides allows them to be effectively internalized by cells. One of the first polycationic peptides can be rightfully considered the Tat peptide, which contains the arginine-rich RKKRRQRRR sequence. A number of studies have been carried out to determine the optimal composition and amount of positively charged amino acid residues. Thus, it was found that, first, peptides rich in Lys, His, or Orn residues are less efficiently absorbed by cells than peptides rich in Arg [162]. This can be rationalized not only by a higher pKa of guanidine groups of arginine (pKa of ca. 13) but also by their ability to form bidentate hydrogen bonds with negatively charged carboxyl, sulfate, and phosphate groups of the compounds present in the cellular membrane, such as phospholipids, acidic polysaccharides, and proteins [163]. Second, the minimum required amount of Arg residues is not less than 6, but to ensure effective cellular uptake, the optimal amount is 8–10 residues [164]. Most of the polycationic CPPs are of natural origin (Tat, penetratin), but synthetic CPPs have also been developed and include arginine homopolymers, peptides of the Pip series developed by the Gait group, and others [52] (more examples in Table 1).

### 4.2. Amphipathic CPPs

The amphipathic class is the most extensive among all CPPs (about 40%) [165]. In addition to positively charged hydrophilic regions, amphipathic peptides also contain hydrophobic regions represented by valine, leucine, isoleucine, and alanine residues [166]. Despite the fact that most amphipathic CPPs are chimeric or synthetic, there are also representatives derived from natural proteins. The amphipathic CPP class is subdivided into three subclasses: primary, secondary, and proline-rich CPPs. Often, primary amphipathic CPPs are chimeric peptides obtained by covalently binding a domain consisting of hydrophobic amino acids (necessary for efficient targeting of cell membranes) with a nuclear localization signal (NLS). An NLS is a short cationic peptide based on lysine, arginine, or proline-rich motives directing peptide conjugates to the cell nucleus through nuclear pores. Representatives of this subclass are MPG peptides [167] and Pep-1 [168], peptides consisting of a hydrophilic part NLS from the large T-antigen of the simian vacuolating virus 40 (SV40) and hydrophobic parts glycoprotein 41 (gp41) of the human immunodeficiency virus (HIV) or a tryptophan-rich cluster, respectively. Natural representatives of this subclass are the ARF (1–22) peptide corresponding to the N-terminal domain of the tumor suppressor protein p14ARF [169], BPrPp (1–28) and MPrPp (1–30) derived from prion proteins [170,171], and others (for more examples, see Table 1). Secondary amphipathic CPPs usually have α-helical conformation with hydrophilic and hydrophobic residues grouped on opposite sides of the helix. Examples of such peptides are the model amphipathic peptide (MAP) [172], transportan [158] or its analogue TP-10 [173], CADY designed by combination aromatic tryptophan and cationic arginine residues [174], and others. It should be noted that among the secondary amphipathic peptides, there are also anionic representatives, such as anionic p28 obtained from azurin [175,176]. The last type of amphipathic peptides is proline-rich CPPs. Due to its secondary amino group, proline cannot serve as a donor of a hydrogen bond for either the α-helix or the β-fold. Such peptides usually form a left-handed polyproline II helix (PPII). An example of proline-rich peptides is a synthetic derivative of Bac 7 (a fragment of antimicrobial protein from the bactenecin family containing 59 amino acids, with four 14-mer repeats); the functions of cell permeability and antimicrobial activity of Bac 7 are concentrated in 24 amino acids (Bac 1–24) [177,178]. Other examples are synthetic proline-rich peptides (PPR)*_n_* and (PRR)*_n_*, where *n* is in the range of 3 to 6 [179].

### 4.3. Hydrophobic CPP

Hydrophobic CPPs consist of non-polar or low-charged amino acid residues and are the smallest class of CPPs. The mechanisms of their cellular penetration are not fully understood but apparently occur due to their high affinity for the hydrophobic domains of cell membranes. Currently, only a limited number of hydrophobic peptides have been found. Examples of hydrophobic CPPs are the C105Y peptide with its C-terminal part of PFVYLI [180] and peptide Pep-7 [181]. More examples of CPPs are presented in Table 1.
molecules-26-05420-t001_Table 1Table 1Most common CPPs used for the delivery of therapeutic nucleic acids.NameSequenceReferencePolycationicTATRKKRRQRRR[182,183,184]pAntRQIKIWFQNRRMKWKKGGC[182,184]PolyarginineR*_n_* (*n* = 8–12)[164](RXR)_4_BRRXRRXRRXRRXRXB[185,186](KFF)_3_KKFFKFFKFFK[187]Pip6aRXRRBRRXRYQFLIRXRBRXRB[188]Pip7bRXRRBRXYRFLIXRBRXRBPip8bRXRRBRXYQFLIRXRRBRBPip9bRXRRBRXFQILYRXRRBRBPip9b2RXRRBRRFQILYRXRXRB**Amphipathic**MPGKETWWETWWTEWSQPKKRK[167]Pep-1GLAFLGFLGAAGSTMGAWSQPKKKRK[168]ARF (1–22)MVRRFLVTLRIRRACGPPRVR[169]BPrPp (1–28)MVKSKIGSWILVLFVAMWSDVGLCKKRPKP[170]MPrPp (1–30)MANLGYWLLALFVTMWTDVGLCKKRPK[171]MAPKLALKALKALKAALKLA[172]TransportanGWTLNSAGYLLGKINLKALAALAKKIL[189]TP-10AGYLLGKINLKALAALAKKIL[173]CADYGLWRALWRLLRSLWRLLWRA[174,190]RICKKWLLRWLSRLLRWLARWLG[191]599GLFEAIEGFIENGWEGMIDGWYGGGGRRRRRRRRRK[192,193]p28LSTAADMQGVVTDGMASGLDKDYLKPD[175,176]Bac7RRIRPRPPRLPRPRPRPLPFP[177,178]Proline-rich peptides(PPR)*_n_* or (PRR)*_n_* (*n* = 3–6)[179]**Hydrophobic**C105YPFVYLI[180]Pep-7SDLWEMMMVSLACQ[181]P4LGAQSNF[194]Pept1PLILLRLLRGQF[195]


## 5. Mechanisms of Peptide-Mediated Delivery

The mechanisms of intracellular transport of CPPs are currently the subject of intensive research. Yet, the pathways, which cell-penetrating peptides employ to penetrate into cells, are still not fully understood. Difficulties in our understanding of cellular uptake mechanisms mainly result from varying physicochemical properties, sizes, and concentration dependence of different CPPs and their conjugates [196]. Nevertheless, it became clear that the same CPP may use different pathways to enter the cell, depending on the conditions of the experiment. In addition, a single CPP may use multiple entry pathways at the same time. Internalization modes may be divided into two groups: energy-independent (direct translocation) and energy-dependent (endocytosis) modes. It is believed that direct translocation occurs when CPPs form nanocomplexes with therapeutic nucleic acids (non-covalent strategy) at high peptide concentrations [197,198]. However, most CPPs and their conjugates appear to be taken up by cells via endocytosis [199,200,201].

### 5.1. Direct Translocation

The process of direct translocation as it is independent of energy can occur even at low temperatures and in the presence of inhibitors of endocytosis. It involves several pathways that are initially based on the interaction of a positively charged CPP with negatively charged membrane components and a phospholipid bilayer. It was reported that one of the pathways occurs when CPPs destabilize the membrane by forming toroidal pores in it [202]. In a recent study on internalization of cationic CPPs, a mechanism was proposed that postulates the formation of a pH gradient across the plasmatic membrane. According to this scheme, at high pH, the carboxyl groups of fatty acids in the lipid bilayer bind to the guanidinium groups of the extracellular CPP and mediate the transfer of the CPP through the plasmatic membrane due to the formation of toroidal pores. In contact with the lower cytosolic pH, the fatty acids of the cell membrane release the CPP into the cytosol, and the pores close [203]. Another model of direct translocation via destabilization of the membrane is the so-called carpet-like mechanism [204,205]. This model is characterized by a change in membrane fluidity upon the interaction of positively charged amino acid residues in a basic CPP with negative charges on the membrane surface. The “inverted micelle” mechanism was also proposed, which envisages CPP capture by invagination of the phospholipid bilayer and the formation of inverted micelles encapsulating the peptide [206]. Thus, translocation of a CPP across the cellular membrane occurs inside micelles, which then discharge the peptide into the cytosol.

### 5.2. Endocytosis

Endocytosis is a natural and energy-dependent process of taking up extracellular molecular cargo inside the cell by encapsulation of the cargo in membranous vesicles, endosomes, which occurs in all cell types. Endocytosis is carried out by a variety of pathways, which may be broadly classified as macropinocytosis, endocytosis mediated by clathrin or caveolin, and clathrin/caveolin-independent endocytosis [207]. Which pathway will be predominant in any distinct case depends mainly on the size and physicochemical nature of the molecular cargo [208]. To avoid their eventual degradation in lysosomes, peptide-oligonucleotide conjugates (POCs) must be released into the cytosol from endosomes formed during endocytotic internalization. Then the conjugate must reach the intracellular targets to have a chance to exhibit biological activity. Release from endosomes appears to be the main limiting factor for the efficient intracellular trafficking of POCs [209]. Although there is a lot of work devoted to the study of the mechanisms for endosomal release, the process is still far from being understood. For example, one model suggests that positive CPP charges can interact with negatively charged components of the endosomal membrane [210]. However, there is some evidence that cationic CPPs covalently bound to large cargoes such as nucleic acids are more likely to remain trapped in endosomes [211]. To avoid endosomal entrapment, various strategies have been developed to increase the efficiency of the endosomal release of various CPP conjugates [212,213,214]. In particular, an approach based on the introduction of pH-sensitive domains into the peptide sequence for destabilization of the lipid membrane at acidic pH inside endosomes showed promising results of facilitating the release of CPPs [215]. Another similar method is based on the introduction of histidine fragments into CPPs. The imidazole ring of histidine (pKa in the range of 5.5–6.5 in proteins [216]) becomes protonated at endosomal pH, leading to an increase in the osmotic pressure in the endosome, which results in the rupture of the membrane and release of the content into the cytosol [217,218,219].

It is also necessary to mention the influence of a CPP on the therapeutic activity of its oligonucleotide cargo. In principle, the cargo may be expected to disassemble from CPPs after having been delivered to its intracellular target. This may happen by dissociation of the complex or by cleavage of the conjugate, depending on the type of chemical bond joining the peptide and oligonucleotide together. In practice, this is not a prerequisite for therapeutic activity. Generally, susceptibility to enzymatic degradation determines the stability of the peptide component in biological media. Peptidases and proteases capable of digesting CPPs are present both in the interior of cells, e.g., in the cytosol, endosomes, and lysosomes, as well as in the extracellular milieu. It was reported that degradation of peptides to a larger extent occurred in endosomes, with only minor contribution from cytoplasmic digestion [220]. Accordingly, when developing a therapeutic POC, it is important to balance the peptide’s resistance to degradation for targeted cargo delivery. Currently, there are few works that compare the antisense activity of the oligonucleotide itself with that of its peptide conjugate. The paper [221] investigated the ability of a peptide-PNA conjugate (P-PNA) to downregulate the luciferase gene in comparison with unconjugated PNA. It was shown that PNA itself does not penetrate into cells and exhibits no antisense activity, while P-PNA inhibits luciferase expression by 60%. To determine whether CPP conjugation has any effect on the antisense activity of PNA, both conjugated and unconjugated PNAs were transfected into HeLa cells permeabilized by streptolysin O to show nearly the same inhibitory activity (about 70%). In addition, it was shown that a cleavable bond between PNA and CPPs has no effect on the antisense activity of the conjugate. These results were achieved using the lysosomotropic agent chloroquine, further proving that endosomal release is critical for antisense activity. Of course, the results could not be expected to apply to all conjugates and their targets. This area is still relatively poorly studied and requires additional research.

## 6. Peptide Additives (Non-Covalent) and Peptide Conjugates (Covalent)

In a broad sense, there may be two types of interactions of a CPP with its cargo: non-covalent and covalent strategies. The majority of the extant peptide-mediated cellular delivery methods for therapeutic nucleic acids are based on covalent bond formation between the peptide and the oligonucleotide parts. Many strategies to form a chemical bond, either stable inside cells or biologically cleavable, between a CPP and an oligonucleotide or the analogue have been explored to date using a variety of reagents and methods. The chemical methods for obtaining covalently linked peptide-oligonucleotide conjugates (POCs) will be discussed in detail below.

On the contrary, the non-covalent strategy does not require the formation of any covalent bond and is often achieved by simply mixing the peptide carrier and its oligonucleotide cargo. It is based on a physical interaction (electrostatic or hydrophobic) between CPPs and a nucleic acid derivative. In this case, short primary or secondary amphipathic CPPs consisting of hydrophobic and hydrophilic parts are most often used, e.g., MPG or Pep-1. As a result, nanoparticulate complexes are formed that are able to pass through the cell membrane with high efficiency via endocytosis [222]. In addition, the formation of such nanocomplexes partially protects the nucleic acid from nuclease digestion. The first example of the use of the non-covalent strategy for oligonucleotide delivery was the MPG peptide additive back in 1997 [167]. Since then, this approach has been extended to other CPPs such as Pep-1, Tat, and polyarginine. The main drawback of the non-covalent strategy is polydispersity of the nanocomplexes, i.e., the production of nanoparticles with different sizes and structures. Such nanoparticles in turn will contain varying amounts of the drug, which may complicate the in vivo application.

## 7. Synthetic Approaches

There are two main concepts of the synthesis of peptide-oligonucleotide conjugates. One is the stepwise solid-phase synthesis of the peptide fragment followed by the oligonucleotide fragment, or vice versa, on the same solid support (also called on-line solid-phase synthesis or in-line solid-phase synthesis in earlier publications). The other is the conjugation of separately assembled peptide and oligonucleotide fragments either on the solid phase or, more frequently, in the solution phase post-synthetically.

The first approach usually implies an automated solid-phase assembly of either the peptide or, less frequently, the oligonucleotide fragment first and then the continuation on the same support of the synthesis of the other fragment, oligonucleotide or peptide, attached via a corresponding functionalized linker introduced at the appropriate stage of the synthesis. The whole procedure is carried out without purification and cleavage of the fragments from the support until the end of the assembly of a full-length conjugate. Two sets of protecting groups and suitable protocols for the synthesis of both fragments have to be used. To avoid side reactions, the two sets of protecting groups should be compatible to allow for smooth deprotection and cleavage from the support at the end of the synthesis so as not to damage the potentially sensitive amino acid and nucleotide residues in both peptide and oligonucleotide fragments. This protecting group compatibility is the main requirement for the successful stepwise solid-phase syntheses of POCs containing problematic residues such as arginine.

In the second approach, the fragments are first synthesized separately using the optimized protocols of peptide or oligonucleotide synthesis, respectively, and sometimes isolated and purified in either completely deprotected or (partially) protected form. The requirement here is for the pair of mutually reactive chemical groups to be introduced into the appropriate positions of the peptide and oligonucleotide fragments. Then the peptide and oligonucleotide fragments are joined together by the chemoselective conjugation reaction either in solution or on the solid phase to form a covalent bond with the participation of the corresponding reactive groups in each fragment. Relative advantages and limitations of the two approaches have been summarized in Table 2.

Most often, the peptide attachment site is at the 5′- or 3′-end of the oligonucleotide, but other positions have also been used for conjugation, such as the 2′-position of the ribose ring or the heterocyclic base. Similarly, the N- or C-termini of the peptide have been used as the site of conjugation, as well as the functional groups of the amino acid side chains.

## 8. Stepwise Solid-Phase Synthesis Approach (On-Line or In-Line Synthesis)

There are two principal schemes for the stepwise solid-phase synthesis of POCs adopted from the 1990s, which differ in the structure of linker moieties joining the respective peptide and oligonucleotide parts together (Figure 4). In the first scheme, a branched trifunctional linker is used, which uses the first functionality to reversibly attach to the solid support, whereas the other two functionalities, one being a protected NH_2_ group and the other a protected OH group, are used for successive peptide synthesis and oligonucleotide synthesis, respectively (Figure 4a). The scheme was employed only occasionally for POC synthesis [223]. In the second, more frequently applied, scheme, a bifunctional linker (or a handle) reversibly attached to the solid support through the first functionality carries either an OH group for the synthesis of the oligonucleotide fragment or a NH_2_ group for the synthesis of the peptide fragment (Figure 4b). Thus, at first, either a peptide or an oligonucleotide fragment is assembled, followed by the introduction of a second linker carrying a temporarily protected OH or NH_2_ group for the synthesis of the second fragment, either oligonucleotide or peptide, correspondingly, and then the second fragment is synthesized using the respective chemistry on the same solid support. The schemes in Figure 4 represent the simplest cases of binary POCs containing peptide and oligonucleotide fragments in the ratio of 1:1. To obtain POCs with different peptide-to-oligonucleotide ratios, e.g., 1:2 or 2:1, more complex synthetic schemes have to be designed.

In most cases, the synthesis of the peptide fragment is carried out according to either Fmoc or, rarely, Boc solid-phase chemistry, and the synthesis of the oligonucleotide fragment is performed almost exclusively by the conventional phosphoramidite method. Therefore, one of the main difficulties in the application of this scheme is the poor compatibility of the respective chemistries for the synthesis of the two fragments. For example, the oligonucleotide fragment may be degraded by undergoing depurination under the harshly acidic conditions of the removal of the *t*-butyl-type protecting groups of the amino acid side chains commonly used in Fmoc/*t*-butyl solid-phase peptide synthesis. This makes it impractical to use the protecting groups of the *t*-butyl family, such as Boc, in the synthesis of the peptide fragment in the purine (i.e., A and G)-containing POCs due to the imminent danger of depurination during the standard concentrated trifluoroacetic acid (TFA) deprotection. Conversely, the peptide fragment can also undergo side reactions under the strongly basic conditions of the final deprotection of the acyl-type *N*-protecting groups used in conventional oligonucleotide synthesis.

Thus, the main difficulty is to select two orthogonal sets of protecting groups for the peptide and oligonucleotide fragments, respectively, and optimize the conditions for their final deprotection and cleavage from the solid support at the end of the synthesis that will preserve the integrity of both fragments of the resulting conjugate. Thus, to carry out successful high-yielding assembly of POCs via stepwise solid-phase synthesis, it is necessary to take into account a number of factors, including careful selection of the combinations of protecting groups, conditions for final deprotection and cleavage, appropriate choice of the solid support, the anchoring group, linkers to join the respective fragments, and optimization of methods for amide and phosphate bond formation.

It should be mentioned that in the case of PNA as an oligonucleotide fragment, the problem of the synthesis of peptide-PNA conjugates is considerably simplified. As PNA itself has a pseudo-peptide structure, the in-line synthesis of the conjugates can be performed sequentially on the same support according to standard Boc/benzyl or Fmoc/*t*-butyl protocol, and there is little difference which of the fragments, peptide or oligonucleotide, has to be assembled at first [224,225,226,227].

### 8.1. Solid Support

The choice of a suitable solid support largely determines the ultimate success of any solid-phase synthesis. The most common polymeric carrier for the synthesis of oligonucleotides is a controlled pore glass (CPG), while polystyrene-based supports are better suited for peptide synthesis. In the majority of published works, CPG was used as a solid support for in-line synthesis of POCs [228,229], but not all cases were able to achieve a high yield of the target conjugate. A number of researchers have proposed to use a copolymer of polystyrene with polyethylene glycol (PEG-PS) as a solid support [230,231], while others carried out the assembly of the conjugates on a standard polystyrene crosslinked with 1% divinylbenzene [232]. In particular, in the work of Robles et al. [233], three solid supports for the synthesis of a conjugate containing pentalysine as a peptide fragment were compared: CPG, PEG-PS, and l% polystyrene-co-divinylbenzene. As a result, the authors almost immediately abandoned the use of PEG-PS as the ninhydrin test showed incomplete coupling of lysine residues. Then, analysis of the final product by gel electrophoresis revealed the presence of two main bands in the lane corresponding to the conjugate synthesized on CPG. Thus, the authors concluded that the best result for the synthesis of peptide conjugates is achieved on 1% polystyrene-co-divinylbenzene resin.

In general, the choice of a solid support should be determined by the strategy of the synthesis of the conjugate (start from the oligonucleotide fragment or start from the peptide fragment). Currently, there is a wide range of solid supports available, which are suitable for both oligonucleotide synthesis and peptide synthesis; however, there is no universally applicable support ideal for both. The choice is restricted only by the aims and particulars of research.

### 8.2. Linkers

#### 8.2.1. Bifunctional Linkers

As mentioned above, it is important to introduce linkers, which serve as a bridge between solid support and peptide and oligonucleotide fragments. The linker should be selected so as to provide selective cleavage of the conjugate from the support at the end of the synthesis. Thus, it has to have an anchoring group to reversibly attach to a solid support and a second functionality to ensure peptide or oligonucleotide chain extension. There are two types of bifunctional linkers: those to carry out the initial synthesis of the peptide fragment and those to carry out the initial synthesis of the oligonucleotide fragment.

Generally, the linkers used for common solid-phase peptide synthesis are acid labile and thus unsuitable for further synthesis of the oligonucleotide fragment. Therefore, most linkers employed to assemble POCs contain an ester bond, which is base labile and cleavable by alkaline hydrolysis, aminolysis, or β-elimination (Figure 5a). There are numerous examples in the literature of the application of such linkers, which were cleaved by concentrated ammonia [234,235,236], ethanolamine [237], sodium hydroxide [238], or tetrabutylammonium fluoride (TBAF) [239] treatment. As reported by Haralambidis et al., treating the conjugate linked to the support via the peptide C-terminal ester with concentrated aqueous ammonia resulted in a mixture of the C-terminal amide and carboxylate [240]. To obtain only the carboxylate peptide fragment, TBAF has been used [239,241], but subsequent treatment with ammonia is still necessary for the complete deprotection of the oligonucleotide fragment. To avoid this, the conditions for simultaneous cleavage from the solid support and final deprotection were optimized by Truffert et al. [238]. In this case, the authors used 0.1 M sodium hydroxide at ambient temperature. As a result, the product was cleaved from the support in just 2 h and complete deblocking of the oligonucleotide fragment occurred in 24 h.

If the conjugate synthesis begins with the oligonucleotide fragment, it is often practical to select a linker incorporating the first nucleoside (Figure 5b). The examples are given in [242,243,244]. The final cleavage from the solid support was carried out by conventional concentrated aqueous ammonia treatment.

Furthermore, it is necessary that the first assembled fragment, either peptide or oligonucleotide, contains a functional group for the synthesis of the second fragment. The most published works usually use a hydroxy amino acid in the peptide fragment as the anchor, and then the oligonucleotide fragment is assembled by the phosphoramidite method, but there are some opposite cases [242,243,244]. Otherwise, the peptide is modified with an additional bifunctional linker, which usually contains a protected hydroxyl group and an activated carboxyl group. After its incorporation into the assembled peptide, the oligonucleotide fragment is synthesized (Figure 4b).

A pioneering article published by Haralambidis et al. back in 1987 is one of the first examples of such a synthesis [245]. The authors used *p*-nitrophenyl 3-[6-(4,4′-dimethoxytrityloxy) ethylcarbamoyl] propanoate as a bifunctional linker containing an activated carboxyl group and a protected hydroxyl group (Figure 6). After the completion of the synthesis, a mixture of TFA and 1,2-ethanedithiol (9:1) was used to remove the protecting groups from the peptide. Cleavage from the solid support was carried out by treatment with concentrated aqueous ammonia for 4 h.

A similar approach was used to synthesize POCs carrying fibrin/filaggrin citrullinated peptides to detect anti-citrullinated protein/peptide antibodies (ACPAs) in patients with rheumatoid arthritis using the ELISA test [246].

#### 8.2.2. Trifunctional Linkers

In addition to the bifunctional linkers mentioned above, trifunctional branched linkers were widely employed in practice. These linkers contained both OH and NH_2_ groups for the synthesis of both oligonucleotide and peptide fragments, respectively, which were usually protected by the orthogonal DMTr and Fmoc groups (Figure 7). It makes possible to assemble one fragment of the conjugate, peptide or oligonucleotide at first, and then the other. With a trifunctional linker, usually the peptide synthesis is carried out before the oligonucleotide synthesis (Figure 4a).

A special case of such a branched trifunctional linker is the amino acid lysine [247]. With the carboxyl group reversibly attached to a solid support, the α- and ε-amino groups protected by the Fmoc and Boc groups, respectively, were used to synthesize the conjugate fragments. The peptide fragment is usually assembled first at the ε-amino group of lysine according to the Boc scheme. Then the α-amino group is modified with a suitable bifunctional linker containing a DMTr-protected hydroxyl group, and the oligonucleotide fragment is then assembled by the phosphoramidite method.

Apart from lysine, a number of trifunctional linkers are described in the literature, e.g., 6-amino-2-(hydroxymethyl)hexan-1-ol [223], 3-aminopropane-1,2-diol [248], and others [249,250,251].

## 9. Post-Synthetic Conjugation Approaches

Poor compatibility of peptide and oligonucleotide chemistries is the main problem in the stepwise solid-phase synthesis approach. Separate syntheses of peptide and oligonucleotide parts followed by linking the fragments, either protected or, more often, partially or fully deprotected a priori, by a selective chemical reaction offers to avoid this difficulty. The approach is called the method of post-synthetic conjugation, which can be carried out in solution, most often with fully deprotected fragments (Figure 8a). Otherwise, post-synthetic conjugation called fragment conjugation on the solid phase is carried out when one of the components is still attached to the solid support during the coupling reaction of peptide and oligonucleotide parts (Figure 8b). In the latter case, the fragment in solution may be fully or partially deprotected, while the support-bound component usually remains protected.

To form a covalent bond between the peptide and oligonucleotide fragments, they must contain functional groups that are mutually reactive in a chemoselective way. There are many methods to form a chemical bond between the conjugate fragments. We will consider them below.

### 9.1. Conjugation via Thioether or Disulfide Bonds

The high reactivity of the thiol group is widely used for the synthesis of POCs. Specific binding of peptide and oligonucleotide fragments can occur through the formation of either a thioether or a disulfide bond. In the first case, the formation of the thioether bond can occur in two main ways: via the Michael addition of thiols to maleimides (Figure 9a) or via the nucleophilic substitution of haloacetamides (Figure 9b).

A haloacetyl group is usually introduced into the peptide or oligonucleotide modified with an aminohexyl group using halogenoacetic anhydride treatment [252,253,254]. A maleimido group can be introduced into the peptide or oligonucleotide using a variety of reagents, such as activated esters of β-maleimidopropionic [255,256], 4-maleimidomethylcyclohexanecarboxylic [257,258], ε-maleimidohexanoic [259,260], or 3-maleimidobenzoic [261] acids (Figure 10). The cysteine residue in the peptide often serves as a source of the thiol group [262].

It should be mentioned that maleimido peptides containing Lys residues have lower stability during long-term storage due to the reaction of a maleimide group with ε-amino groups of Lys [263]. To avoid side reactions, the authors recommended to use the modified peptide immediately for the conjugation reaction. Thus, it may be a better option to introduce the maleimide group into the oligonucleotide fragment, although a number of studies adhere to the opposite point of view.

In the case of disulfide-linked conjugates, there are two ways of forming the disulfide bond between peptide and oligonucleotide fragments. The first way (Figure 11a) is the direct oxidation of the two fragments, each containing a thiol group [264]. The second way consists of the modification of one of the thiol-containing fragments to form an activated disulfide, most often with a pyridylsulfenyl (Pys) [265,266] or a 3-nitropyridylsulfenyl (Npys) group (Figure 11b) [267,268].

A significant disadvantage of the first way is poor selectivity. Homodimers can form from peptide or oligonucleotide fragments as by-products in the reaction. The use of the activating groups such as Pys or Npys provides the necessary selectivity of the conjugation and increases the yield of the conjugate. According to the comparison conducted in the [269], oligonucleotide activation results in the highest conjugation yields. Moreover, the activated pyridyl disulfides are base labile and thus problematic to use during solid-phase oligonucleotide synthesis because of the final deprotection with concentrated aqueous ammonia. Therefore, conjugation in solution seems to be the most appropriate scheme for coupling of peptides and oligonucleotides through the disulfide bond.

Recently, another method for the synthesis of POCs with a disulfide bond using *S*-sulfonate-protected cysteine of the peptide was developed. *S*-sulfonates undergo thiolysis to form disulfide-linked conjugates with free thiol compounds. The thiol group was introduced into the oligonucleotide through the 2′-position, followed by attachment of the nucleoside to a solid support. The method has been optimized for both conjugation in solution [270] and fragment conjugation on the solid phase [271].

### 9.2. Conjugation through Oxime, Thiazolidine, or Hydrazone Bonds

The use of oxime, thiazolidine, and hydrazine groups to form a covalent bond between peptide and oligonucleotide fragments is widely used for conjugation. The reaction takes place under mild conditions and uses highly reactive functional groups. Namely, carbonyl compounds, such as aldehydes, especially glyoxylic acid derivatives or, more rarely, ketones, react with compounds containing aminooxy groups, 1,2-aminothiol groups (usually coming from cysteine), and hydrazine or hydrazide groups to form *O*-alkyl oximes, thiazolidines, and hydrazones, respectively (Figure 12). It should be noted that an oligonucleotide equipped with a carbonyl group and a peptide with, e.g., an aminooxy group, are more preferable to couple because aminooxy oligonucleotides tend to react with traces of aldehydes and ketones, such as acetaldehyde and acetone, present in solvents. In the case of thiazolidine formation, the reaction is better to be carried out under oxygen-free conditions due to the risk of oxidation of a free thiol group.

Several phosphoramidite reagents for the introduction of masked aldehyde precursors, such as 1,2-aminoalcohol (Figure 13a), 1,2-diol (Figure 13b), or acetal (Figure 13c), onto the 5′-end of an oligonucleotide during solid-phase synthesis have been developed [272,273,274]. The release of glyoxylic acid amide or aliphatic or aromatic aldehyde groups was carried out after the end of oligonucleotide synthesis after usual ammonia deprotection by treatment with acetic acid, followed by periodate ion oxidation (Figure 13a,b).

A similar strategy for the introduction of an aldehyde or a glyoxylic amide onto the 3′-end of an oligonucleotide was developed. Commercially available CPG supports (Figure 14a,b) can be employed to generate an aldehyde group after periodate cleavage of the corresponding 1,2-diol or 1,2-aminoalcohol [275,276]. Another solid support (Figure 14c) was obtained from commercially available LCAA-CPG and *N^α^*-Fmoc-*O*-*t*-Bu-serine in several steps and used to produce a glyoxylic acid amide upon periodate oxidation [277]. Oligonucleotides having aldehyde or glyoxylic acid amide groups were obtained in the same way.

Such derivatives of oligonucleotides containing aldehyde or glyoxylic acid amide groups at the 3′- or 5′-ends employed to obtain POCs in good yields via oxime, thiazolidine, or hydrazone formation have been described in [274,276,278,279,280,281], respectively.

Moreover, oligonucleotide derivatives containing reactive carbonyl groups at both 3′- and 5′-ends have been obtained in the same way. Subsequent addition of aminooxy peptides to such bifunctional oligonucleotide derivatives furnished the respective 3′,5′-bis-conjugates through oxime bond formation [282]. There is a limitation to this approach as only bis-conjugates with the same peptide can be obtained from bis-aldehyde-containing oligonucleotides. However, the same method could be modified to produce conjugates with two different peptides or with a peptide and a label, e.g., a fluorophore [283].

Recently, conjugation through an oxime bond was reported for 5-formyl-dC or 7-(2-oxoethyl)-7-deaza-dG and a peptide containing unnatural oxylysine amino acid [284]. This method makes it possible to conjugate peptides at the internal nucleobase position within the oligonucleotide fragment, leaving the 3′- and 5′-end free.

In addition to oligonucleotides modified with carbonyl groups, phosphoramidite reagents for the introduction of hydrazide or aminooxy groups into oligonucleotides via solid-phase synthesis have been developed [285]. Such oligonucleotides could also be applied for conjugation to peptides (Figure 15).

POCs conjugated at the 2′-position of the ribose residue through the incorporation of a suitably modified nucleoside have also been prepared [286,287,288]. An advantage of 2′-conjugation is that it leaves both 5′- and 3′-ends of the oligonucleotide free to attach other groups, such as fluorescent or radioactive labels. One of the examples involves the use of 2′-*O*-(2,3-dibenzoyloxypropyl)-rU phosphoramidite. Oligonucleotides containing the 2′-*O*-β-oxoethyl group were obtained after ammonia deprotection removing the benzoyl groups, followed by periodate oxidation of the corresponding 2′-*O*-(2,3-dihydroxypropyl)-rU precursor [289]. The 2′-conjugates linked by oxime, thiazolidine, and hydrazine bonds were successfully obtained, and the latter were produced by sodium cyanoborohydride reduction of the corresponding hydrazones, which were found to be sensitive to hydrolysis.

An elegant method of obtaining 2′-conjugates via *N*-methoxyoxazolidine formation was developed recently [290]. Unusually, this approach employed a peptide with an aldehyde group for conjugation to an oligonucleotide containing a 2′-*N*-methoxyamino group and a free 3′-OH. The conjugate decomposed into peptide and oligonucleotide fragments under slightly acidic conditions, displaying negligible decay at pH 7. Such POCs could fall apart at acidic pH and release their cargo after going inside cells via endocytosis.

### 9.3. Conjugation through Amide Bonds

One of intrinsically selective ways to form an amide bond is the native chemical ligation approach of Dawson and Kent. Originally, the method was developed for the condensation of a fully unprotected synthetic peptide C-terminal thioester and the peptide containing an N-terminal cysteine residue [291]. Stetsenko and Gait adapted the native ligation method for the synthesis of POCs [292,293]. In this method, a 5′-modified oligonucleotide incorporating a cysteine residue with the thiol group masked by *t*-butyl disulfide and a peptide with an N-terminal thioester are synthesized separately on their own solid supports, cleaved and deprotected, and isolated and purified, if necessary, and then conjugated in solution after reductive removal of the *t*-butyl disulfide (Figure 16). Later, Cys-containing uridine phosphoramidite was developed by Diezmann et al. to carry out internal 2′-conjugation with peptides by native chemical ligation [294].

Recently, a site-specific peptide-oligonucleotide conjugation method involving the oxanine nucleobase and the N-terminal Cys residue in a peptide was proposed. As a result of intramolecular rearrangement after nucleophilic attack by the thiol group, it was possible to obtain a conjugate with the peptide located anywhere within an oligonucleotide chain [295]. A disadvantage of the above method is a possible disruption of complementary base pairing.

In addition to native ligation, a convenient method of conjugation through the amide bond mediated by a peptide-coupling reagent, such as HBTU, was developed for oligonucleotides modified with a 5′-terminal carboxyl group. Kachalova et al. described a non-nucleosidic phosphoramidite building block, which has the carboxylic acid moiety masked by the acid-labile 2-chlorotrityl group. First, the oligonucleotide fragment, while still protected and attached to the solid support, was detritylated under usual mildly acidic conditions to unmask the carboxyl group, followed by activation with a suitable peptide-coupling reagent, such as HBTU/HOBt/DMF. Then the conjugation was carried out by adding an amine or a short peptide with a free N-terminus to the oligonucleotide immobilized on the solid support [296]. The phosphoramidite is now commercially available from major suppliers, such as 5′-carboxy-modifier C5. Later, the method was optimized for the synthesis of 2′-conjugates through the formation of amide bonds either on the solid phase or in solution [297,298].

An example of an opposite approach employs an oligonucleotide modified with a thymidine residue having an amino group at the *C*-5 position [299]. Conjugation was carried out with the C-termini of various amino acids and dipeptides. In this case, water-soluble 1-ethyl-3-(3-dimethylaminopropyl)carbodiimide was used as an activating reagent for the carboxyl group. A drawback of this method is the danger of racemization of the peptide.

In a number of studies, phosphordiamidate morpholino oligomers (PMOs) were employed as oligonucleotide components. It was found that the conjugation of cell-penetrating peptides containing multiple arginine residues to charge-neutral PMOs (see Figure 2, **7**) is usually more straightforward than to negatively charged oligonucleotides due to the absence of ionic interaction for PMOs. In this case, the C-terminus of the peptide ending in an achiral amino acid, such as β-alanine or ε-aminohexanoic acid, was activated by a mixture of HBTU/HOBt/DIEA under non-aqueous conditions and then conjugated to the 3′-terminal NH group of the PMO [300,301].

### 9.4. Conjugation through Click Chemistry (1,3-Dipolar Cycloaddition Reaction of Alkynes to Azides)

The reaction between azides and alkynes has been known for a long time [302,303]. However, when the Meldal and Sharpless groups independently reported the copper(I)-catalyzed variant of the reaction [304,305], it was hailed as a golden standard of click chemistry [306]. Up to now, the 1,3-dipolar cycloaddition of terminal alkynes to azides has been widely employed to prepare peptide-oligonucleotide conjugates.

Oligonucleotides can be readily functionalized with terminal alkyne residues by means of special phosphoramidite modifiers containing an alkynyl group, such as 5′-*O*-propynyl-*N*^3^-benzoyl-dT phosphoramidite used by Gogoi et al. (Figure 17) [307]. On the contrary, PNAs or PMOs are more frequently functionalized by the azido group using special derivatives, such as α-Fmoc-ε-azido-L-lysine [308]. In an interesting adaptation of this reaction for the attachment of the peptide fragment at the internal position of an oligonucleotide, Astakhova et al. synthesized a 21-mer oligonucleotide containing single or double internal 2′-alkynyl-LNA nucleotides, and then conjugation was carried out with azide derivatives of peptides [309].

Phosphorothioate (PS) oligonucleotides (Figure 2, **1b**) represent one of the most commonly used type of modified DNA analogues due to their increased resistance to nuclease digestion and favorable pharmacokinetics. However, until recently, there were almost no examples of conjugation reactions with phosphorothioates by the Cu(I)-catalyzed azide-alkyne cycloaddition, which was associated with the adverse influence of copper ions on the stability of the PS bond, giving rise to the impression that this type of click chemistry is incompatible with phosphorothioate oligonucleotides. The Strömberg group designed an optimized alkyne-azide cycloaddition protocol for the high-yielding synthesis of phosphorothioate conjugates [310]. The reaction was carried out by the fragment coupling method on the solid phase using either commercial or synthesized in-house PS oligonucleotides, easily obtainable linkers, and the copper (I) bromide-dimethyl sulfide complex as a catalyst.

The peptide fragments can also be functionalized with either an azide or an alkyne during solid-phase synthesis using, e.g., Boc-(2*S*,4*S*)-4-azidoproline to introduce the azido group and propynoic acid or Fmoc-l-β-homopropargylglycine to obtain the alkynyl derivative.

One of the advantages of this type of conjugation is the possibility of carrying out the reaction both in aqueous and in organic solvents. In addition, 1,3-dipolar cycloaddition between alkynes and azides is intrinsically chemoselective, which allows for conjugation of the fragments without the need for any protecting groups. It was also found that, in contrast to the thiol-maleimide conjugation, when the solubility of the peptide strongly influences the reaction rate, the copper-catalyzed alkyne-azide cycloaddition proceeds well even with sparingly soluble peptides [311].

### 9.5. Conjugation through the Diels-Alder Reaction

The Diels-Alder and inverse electron-demand Diels-Alder reactions are a convenient and increasingly popular method for bioconjugation of various molecules as they can occur in aqueous media with high yield and chemoselectivity. The Diels-Alder reaction in general is a [4 + 2] cycloaddition occurring between a 1,3-diene and an unsaturated compound—the dienophile. Typically, dienes contain electron-donating and dienophiles contain electron-withdrawing substituents. Less common is the inverse variant of the reaction, when the dienophile is electron rich and the diene is electron poor.

Grandas et al. described the application of Diels-Alder cycloaddition for the preparation of peptide-oligonucleotide conjugates [312]. The conjugates were obtained by the reaction between an oligonucleotide derivatized at the 5′-end by an acyclic diene and a maleimido peptide (Figure 18). The cycloaddition was carried out under mild conditions in aqueous solution at 37 °C. The speed of the reaction was found to vary depending on the size of the reagents, but it can be completed in 8–10 h by treating the diene-modified oligonucleotide with a small excess of the maleimido peptide.

As maleimide is not stable to the ammonia deprotection of oligonucleotides, the maleimide moiety attached to the peptide was more often used as a dienophile in conjugation reactions until the Grandas group developed a clever method for the introduction of the maleimide group into oligonucleotides [313]. In this approach, 2,5-dimethylfuran was exploited as a protecting group for maleimide removable by heat-promoted retro-Diels-Alder reaction without affecting the oligonucleotide. Moreover, it was shown that simultaneous deprotection and conjugation provided a faster reaction and better yields.

Recently, inverse electron-demand Diels-Alder reaction has been applied for the preparation of POCs [314]. The authors used 7-oxanorbornene as a dienophile and tetrazine as a diene. The 7-oxanorbornenes were synthesized by Diels-Alder reactions between maleimides and furans. The oligonucleotide or peptide fragments can be obtained using special oxanorbornene-containing phosphoramidite or carboxylic acid, respectively (Figure 19). The conjugates were produced in good yields with a low amount of by-products.

The methods described above of course are not exhaustive. Many more ingenious methods for post-synthetic conjugation of oligonucleotides to peptides have been developed over more than three decades of research, but nowadays many of these are only rarely referred to in the literature [315,316,317].

## 10. Comparison of the Two Approaches: Conclusions

The undoubted advantage of the stepwise solid-phase synthesis approach is the absence of time-consuming isolation and purification of the individual peptide and oligonucleotide fragments of the conjugate. When the stepwise yields are sufficiently high, the final product requires only a single purification procedure, usually chromatography. Yet, this approach has obvious disadvantages arising from limited compatibility of the chemistries used for the synthesis of peptide and oligonucleotide fragments of the conjugate, notwithtsanding the side reactions during the deprotection of both. As there is only a limited number of amino acids that can be attached without the use of protective groups, judicious choice of the latter is required; the most difficult case remains arginine. Another restriction is the increasing difficulty of obtaining longer than medium-length conjugates because of the number of steps required in solid-phase synthesis and the need to maintain as high yield as possible on each step; this likely leaves out of question any of the potential “difficult sequences”.

In turn, the second approach also has a number of disadvantages. The post-synthetic conjugation involves a number of prior steps. First, it is necessary to complete the solid-phase assembly, complete or, sometimes, partial deprotection, and, most often, purification of the two fragments and then carry out the synthesis, isolation, and purification of the conjugate. That may result in significant losses in the isolated yield of the final product. In addition, predominantly in the case of a cationic or a highly hydrophobic peptide, there may be serious problems with its solubility and the solubility of the resulting conjugate due to possible aggregation and precipitation. Thus, charge-neutral oligonucleotide analogues, such as PNA or PMO, are better suited for peptide conjugation in solution. However, despite all the disadvantages, the second approach is currently a favored and much more frequently exploited method, not the least because of the availability of many excellent conjugation chemistries, such as aldehyde and oxime/hydrazone, alkyne and azide, and Diels-Alder reactions. It allows one to avoid a painstaking selection of synthetic conditions necessary for in-line synthesis. That is why, with the exception of a handful of papers published some years ago [235,246], at present there are almost no new examples of the application of the stepwise solid-phase synthesis approach for the preparation of POCs in the literature. However, it should be noted that in the case of PNA, as mentioned earlier, in-line synthesis could still be the method of choice. Nevertheless, the post-synthetic conjugation approach looks more attractive today, although it is also not without drawbacks and limitations.

Unfortunately, there is no universal method for the synthesis of peptide-oligonucleotide conjugates, and we probably should not expect the one to come due to the exceeding variety in the physicochemical properties of cell-penetrating peptides (see Table 1). Thus, in the majority of cases, the choice of the method and conditions for the synthesis of POCs have be determined individually in each specific instance. However, the expanding therapeutic potential of oligonucleotides and the advantages of their targeted delivery by conjugation to peptides lead to the continuing search for new and more convenient methods for the preparation of their conjugates.

## Figures and Tables

**Figure 1 molecules-26-05420-f001:**
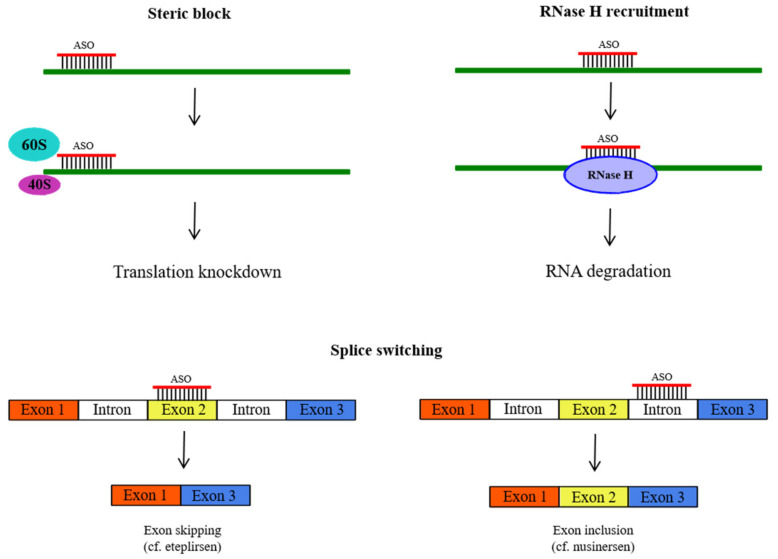
The aspects of the antisense mechanism.

**Figure 2 molecules-26-05420-f002:**
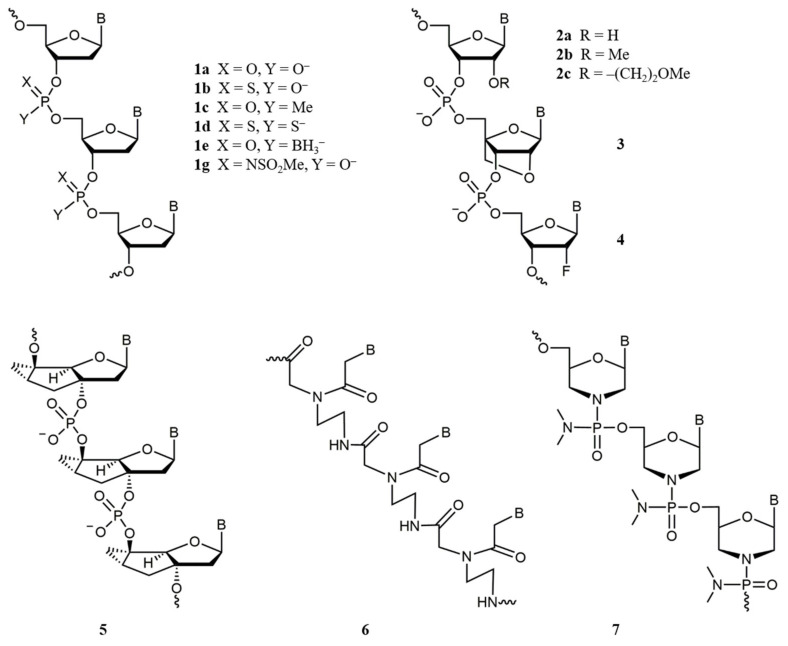
Oligonucleotides and their analogs: (**1a**) native DNA, (**1b**) phosphorothioate, (**1c**) methyl phosphonate, (**1d**) phosphorodithioate, (**1e**) boranophosphate, (**1f**) mesyl phosphoramidate, (**2a**) native RNA, (**2b**) 2′-*O*-methyl RNA, (**2c**) 2′-*O*-(2-methoxy)ethyl RNA, (**3**) bridged/locked nucleic acid (B/LNA), (**4**) 2′-α-fluoro DNA, (**5**) tricyclo-DNA (tcDNA), (**6**) peptide nucleic acid (PNA), and (**7**) phosphordiamidate morpholino oligomer (PMO).

**Figure 3 molecules-26-05420-f003:**
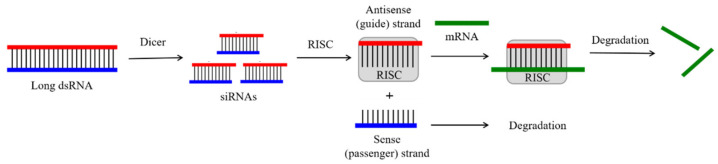
The mechanism of RNA interference (RNAi) mediated by small interfering RNAs (siRNAs).

**Figure 4 molecules-26-05420-f004:**
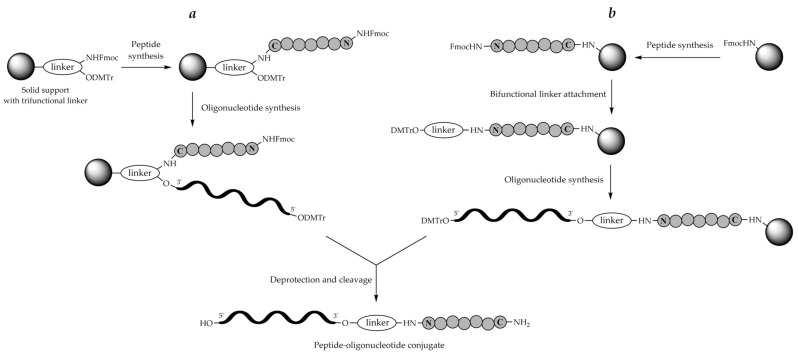
Two main schemes for stepwise solid-phase synthesis of POCs: (**a**) with a branched trifunctional linker and (**b**) with a bifunctional linker. Note: The oligonucleotide fragment can be assembled first, followed by the peptide fragment. DMTr—4,4′-dimethoxytrityl group; Fmoc—9-fluorenylmethoxycarbonyl group; C and N—peptidic C- and N-termini, respectively.

**Figure 5 molecules-26-05420-f005:**
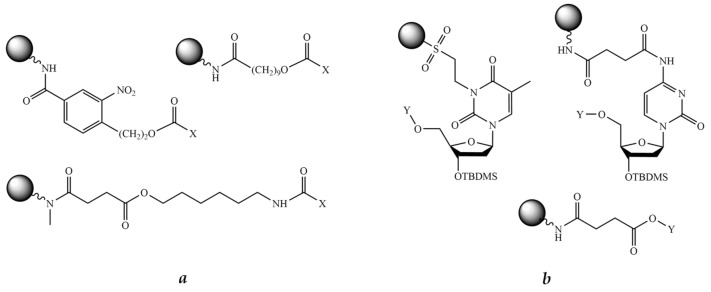
Bifunctional linkers used in the stepwise solid-phase synthesis approach: (**a**) starting with a peptide fragment via the *C*-terminus (X) and (**b**) starting with an oligonucleotide fragment (Y). TBDMS—*t*-butyldimethylsilyl protecting group.

**Figure 6 molecules-26-05420-f006:**
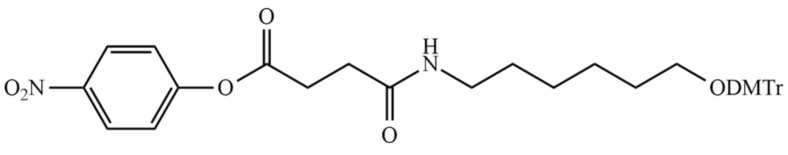
Example of a bifunctional linker connecting peptide and oligonucleotide fragments.

**Figure 7 molecules-26-05420-f007:**
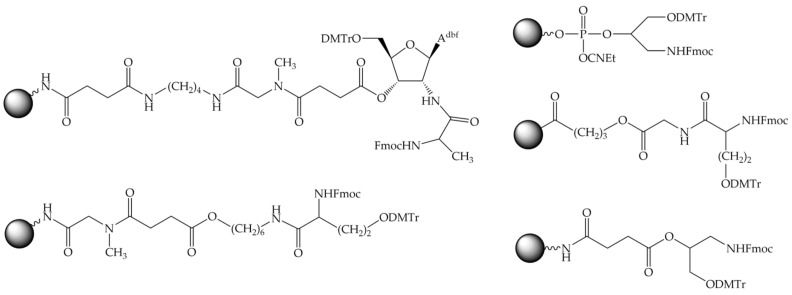
Trifunctional linkers used in the stepwise solid-phase synthesis approach. CNEt—2-cyanoethyl; dbf—di-*N*,*N*-butylformamidine.

**Figure 8 molecules-26-05420-f008:**
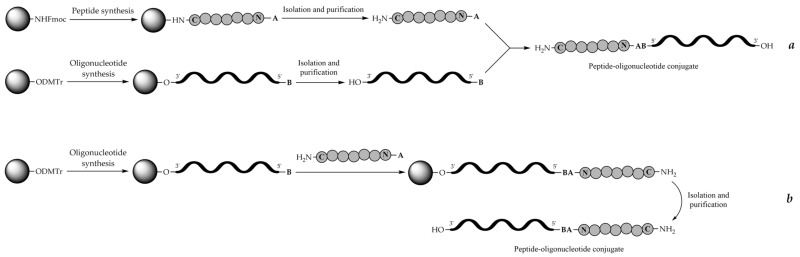
The main schemes of the post-synthetic conjugation approach: (**a**) conjugation in solution and (**b**) fragment conjugation on the solid phase.

**Figure 9 molecules-26-05420-f009:**
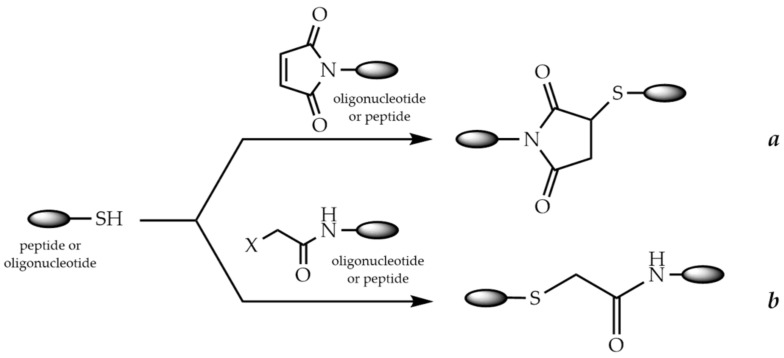
Conjugation through the thioether bond: (**a**) Michael addition of thiols to maleimides and (**b**) nucleophilic substitution of haloacetamides; X = I, Br, or Cl.

**Figure 10 molecules-26-05420-f010:**
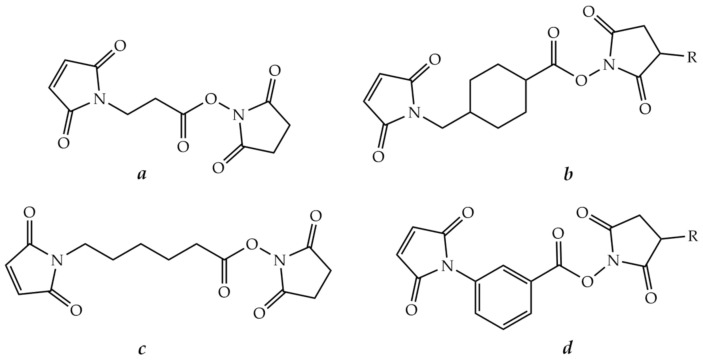
Reagents for the introduction of the malemide group: activated *N*-succinimidyl esters of β-maleimidopropionic (**a**), 4-maleimidomethylcyclohexanecarboxylic (**b**), ε-maleimidohexanoic (**c**), and 3-maleimidobenzoic (**d**) acids; R = H or SO_3_Na.

**Figure 11 molecules-26-05420-f011:**
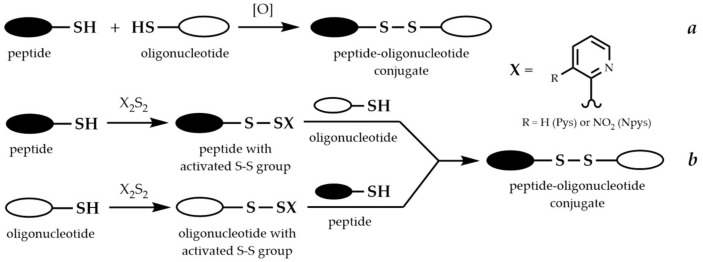
Conjugation through the disulfide bond: (**a**) by direct oxidation and (**b**) via activation by a pyridylsulfenyl (Pys) or a 3-nitropyridylsulfenyl (Npys) group.

**Figure 12 molecules-26-05420-f012:**
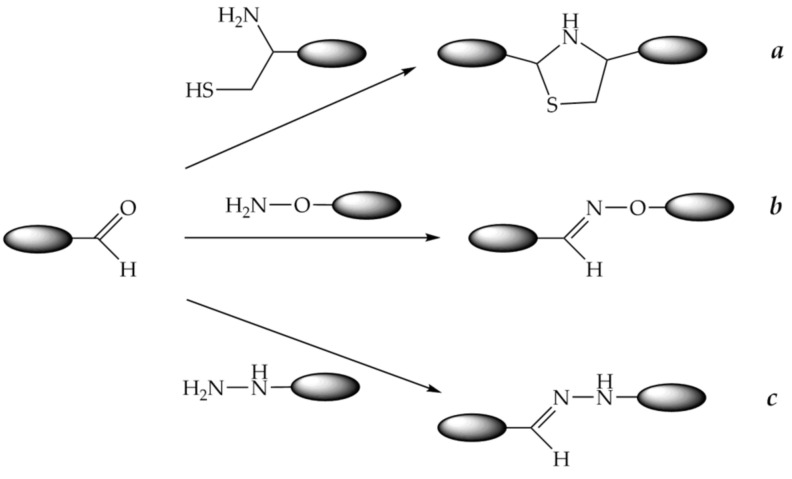
Conjugation through thiazolidine (**a**), oxime (**b**), and hydrazone (**c**) bonds, respectively.

**Figure 13 molecules-26-05420-f013:**
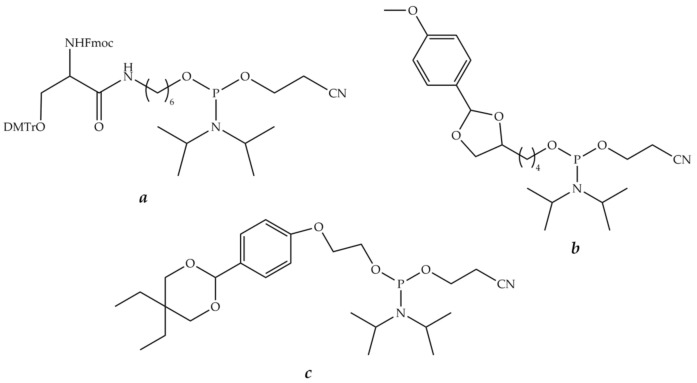
Masked phosphoramidite derivatives for the introduction of glyoxylic acid amide (**a**) and aldehyde (**b**,**c**) groups into oligonucleotides at the 5′-end.

**Figure 14 molecules-26-05420-f014:**
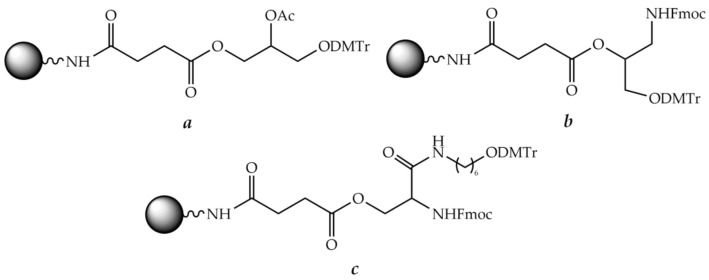
CPG supports for the introduction of aldehyde (**a**,**b**) or glyoxylic acid amide (**c**) groups at the 3′-end of an oligonucleotide.

**Figure 15 molecules-26-05420-f015:**
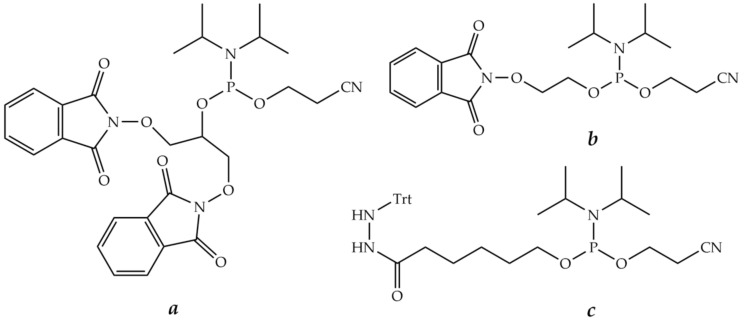
Phosphoramidite reagents for the introduction of aminooxy (**a**,**b**) and hydrazide (**c**) groups into oligonucleotides. Trt—triphenylmethyl (trityl) group.

**Figure 16 molecules-26-05420-f016:**
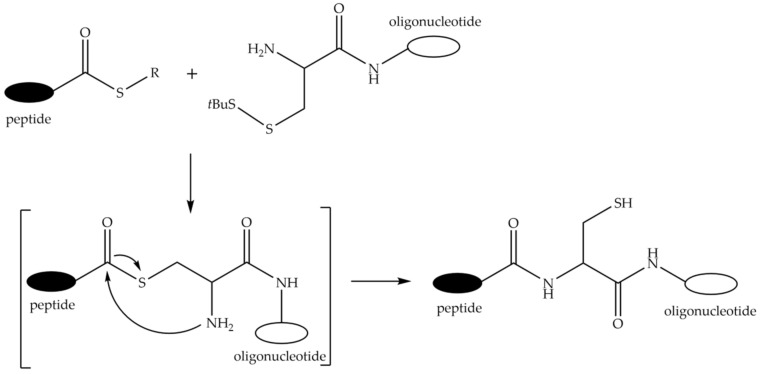
Synthesis of peptide-oligonucleotide conjugates via native chemical ligation. R—benzyl [292].

**Figure 17 molecules-26-05420-f017:**
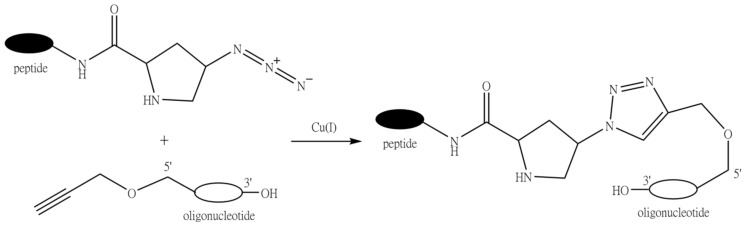
An example of peptide-oligonucleotide conjugation via the copper(I)-catalyzed 1,3-dipolar cycloaddition reaction of alkynes to azides [307].

**Figure 18 molecules-26-05420-f018:**
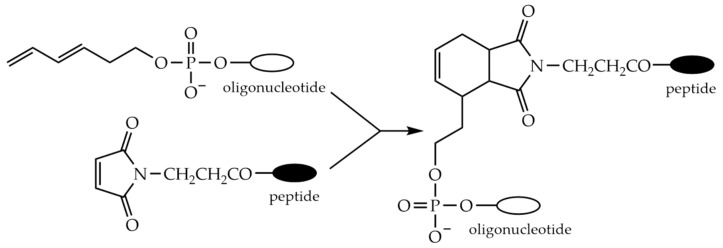
Conjugation through Diels-Alder reaction.

**Figure 19 molecules-26-05420-f019:**
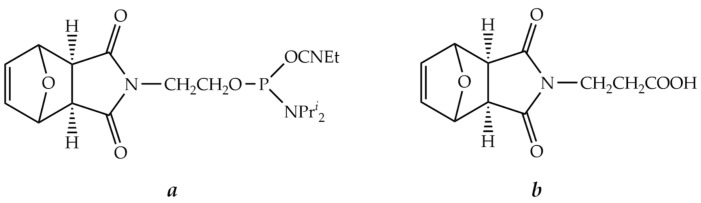
Oxanorbornene derivatives for the introduction of a masked maleimide into oligonucleotides (**a**) or peptides (**b**). CNEt—β-cyanoethyl group.

**Table 2 molecules-26-05420-t002:** Main approaches to the synthesis of POCs, and their advantages and limitations.

Stepwise Solid-Phase Synthesis		
Conjugation via	Advantages	Disadvantages/Limitations
Bifunctional or trifunctional linker	Absence of time-consuming isolation/purification of both peptide (P) and oligonucleotide (O) fragments No excess of either P or O fragment—less solubility problems May be convenient for peptide-PNA conjugates (P-PNAs) due to protecting group compatibility	Poor compatibility of P and O chemistries: the need to design a suitable protecting group scheme. Attachment of limited number of amino acids without side-chain protection Difficulty synthesizing longer than medium-length conjugates
**Post-Synthetic Conjugation**		
**Conjugation via**	**Advantages**	**Disadvantages/Limitations**
Thioether or disulfide bond	Many suitable conjugation procedures available Many reagents for functionalization of either fragment available No problem with incompatibility of the two chemistries Conjugation of peptides with any amino acid composition Conjugation of peptides of almost any length (up to proteins)	Separate multistep preparation and purification of both fragments Reaction in aqueous solvents Solubility problems with polycationic or highly hydrophobic peptides
Native ligation		
Oxime, thiazolidine, or hydrazone linkage		
Amide bond formation		
Click chemistry		
Diels-Alder reaction		

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
