# Peer review of "Chemistry of Peptide-Oligonucleotide Conjugates: A Review"

_molecules, 2021, doi:10.3390/molecules26175420_

Round 1
Reviewer 1 Report
The authors provide an overview of both the motivation and chemical pathways to preparing peptide-oligonucleotide conjugates (POCs). An extensive number of references are included and the overall organization of the manuscript is meaningful, especially given the challenge of providing a written description of so many chemical routes. Overall, this reviewer has a favorable impression of how the authors covered the breadth and range of this challenging field while also striving to include helpful details that clarify limitations with these various chemical routes. It is also appreciated that the authors give some motivation and background in the first several sections (e.g. nucleic acid therapeutics) before delving into a lot of detailed chemistry descriptions. To better help the reader navigate the various sub-topics (starting with section 7) that then follow, my first suggestion below is likely the most pertinent editing suggestion.
1. While several figures are included throughout the manuscript to highlight the chemical pathways, the addition of a summary table could be particularly helpful, especially given the manuscript’s length. The table could be broken into two main sections (1) stepwise solid-phase synthesis of POCs and (2) post- synthesis conjugation of oligonucleotide and peptide fragments. Under each of these two sections, the authors could list the various chemical routes and include key advantages and disadvantages/limitations that are mentioned throughout each subsection (e.g. section 9.1 conjugation via thioether; conjugation via disulfide bond) Note: while the reviewer understands that the authors intend to clarify the completion of POC is achieved in the first method above, it is suggested that the authors remove the word “total” from the phrase “total stepwise solid-phase synthesis” throughout the manuscript (e.g. Figure 8 caption). It is also suggested to replace the term “synthetic” with the term “synthesis” in the phrase “post-synthetic conjugation” that is also used throughout the manuscript (e.g. section 9 header).
2. My next major suggestion asks the authors to reflect on some additional practical implication of POCs. It appears that the key motivation for the POC field is to enable nucleic acid therapeutics to cross the cell membrane barrier. So, once the peptide has carried out this major task, can the authors reflect on the potential advantages and disadvantages to the peptide remaining as part of the cargo? Would these peptides, for example, likely interfere with subsequent antisense activities of the nucleic acids?
The remaining suggestions below are fairly minor, but intended to help the authors improve the narrative flow of their manuscript by amending some wording and stylistic writing choices.
3. The authors overuse commas, articles (e.g. the) and prepositional phrases throughout the manuscript resulting in overly long or wordy phrases. For example, it is suggested that the following sentence (p.1, paragraph 1) below…
“As peptide-oligonucleotide conjugates are, thus, chimeric compounds, which include an (oligo)peptide part and a nucleic acid part, their properties are a combination of the properties of the parent bio-molecules such as the immanent ability of nucleic acids for the complementary recognition and multifaceted properties of much more structurally and functionally diverse pep-tides.”
…can be simplified as illustrated below…
“As chimeric compounds which include an (oligo)peptide part and a nucleic acid part, each peptide-oligonucleotide conjugate (POC) represents a combination of its parent bio-molecules such as the immanent base-pairing ability of nucleic acids and the multifaceted bioactivity of the structurally and functionally diverse peptides.”
Though this reviewer has only provided one example, this editing suggestion is not likely to be trivial to execute.
4. In Figure 2, the authors should include the names of conjugates 3-7 in either the figure itself or in the caption.
Author Response
Responses to Reviewers’ comments
Reviewer 1
- “While several figures are included throughout the manuscript to highlight the chemical pathways, the addition of a summary table could be particularly helpful, especially given the manuscript’s length. The table could be broken into two main sections (1) stepwise solid-phase synthesis of POCs and (2) post- synthesis conjugation of oligonucleotide and peptide fragments. Under each of these two sections, the authors could list the various chemical routes and include key advantages and disadvantages/limitations that are mentioned throughout each subsection (e.g. section 9.1 conjugation via thioether; conjugation via disulfide bond).”
We have inserted a new Table 2 (page 12) divided into two sections providing the summary Reviewer 1 outlined.
“Note: while the reviewer understands that the authors intend to clarify the completion of POC is achieved in the first method above, it is suggested that the authors remove the word “total” from the phrase “total stepwise solid-phase synthesis” throughout the manuscript (e.g. Figure 8 caption). It is also suggested to replace the term “synthetic” with the term “synthesis” in the phrase “post-synthetic conjugation” that is also used throughout the manuscript (e.g. section 9 header).”
We have deleted the word “total” from the term ‘total stepwise solid-phase synthesis” as Reviewer 1 suggested. However, we would prefer to stick to the term “post-synthetic conjugation” rather than “post-synthesis conjugation” as we believe the former is almost three times more common than the later as revealed by simple Google search (cf. https://www.google.ru/search?q=post-synthetic&newwindow=1&client=opera&ei=vW8jYcyLLOyOxc8Pv8SusAM&oq=post-synthetic&gs_lcp=Cgdnd3Mtd2l6EAwyBAgAEBMyBAgAEBMyBAgAEBMyBAgAEBMyBAgAEBMyBggAEB4QEzIGCAAQHhATMgYIABAeEBMyBggAEB4QEzIGCAAQHhATOgcIABBHELADOggIABAKEB4QE0oECEEYAFCvNli2Q2DdbWgBcAJ4AIABkAGIAcEHkgEDMC43mAEAoAEByAEIwAEB&sclient=gws-wiz&ved=0ahUKEwiM3qK-78byAhVsR_EDHT-iCzYQ4dUDCA0 vs https://www.google.ru/search?q=post-synthesis&newwindow=1&client=opera&source=hp&ei=-W8jYZrqBcvB5OUP8Kqa8AM&iflsig=AINFCbYAAAAAYSN-CfI6svTD6O32jwCH5m9O1J_1z7E0&oq=post-synthesis&gs_lcp=Cgdnd3Mtd2l6EAEYADIECAAQEzIECAAQEzIGCAAQHhATMgYIABAeEBMyBggAEB4QEzIGCAAQHhATMgYIABAeEBMyBggAEB4QEzIGCAAQHhATMggIABAKEB4QEzoFCAAQgAQ6CwguEIAEEMcBENEDOgsILhCABBDHARCjAjoFCC4QgAQ6DgguEIAEEMcBENEDEJMCOg4ILhCABBDHARCjAhCTAjoHCAAQyQMQQzoECAAQQzoECC4QQzoLCC4QgAQQxwEQrwE6BwgAEIAEEAo6BAgAEAo6BggAEA0QHlDADVjoLmCaQWgAcAB4AIABqAGIAf8OkgEEMC4xNJgBAKABAQ&sclient=gws-wiz).
- “My next major suggestion asks the authors to reflect on some additional practical implication of POCs. It appears that the key motivation for the POC field is to enable nucleic acid therapeutics to cross the cell membrane barrier. So, once the peptide has carried out this major task, can the authors reflect on the potential advantages and disadvantages to the peptide remaining as part of the cargo? Would these peptides, for example, likely interfere with subsequent antisense activities of the nucleic acids?”
We added a paragraph discussing this question to the section 5.2 (pages 10 and 11).
- “The authors overuse commas, articles (e.g. the) and prepositional phrases throughout the manuscript resulting in overly long or wordy phrases. For example, it is suggested that the following sentence (p.1, paragraph 1) below…
“As peptide-oligonucleotide conjugates are, thus, chimeric compounds, which include an (oligo)peptide part and a nucleic acid part, their properties are a combination of the properties of the parent bio-molecules such as the immanent ability of nucleic acids for the complementary recognition and multifaceted properties of much more structurally and functionally diverse pep-tides.”
…can be simplified as illustrated below…
“As chimeric compounds which include an (oligo)peptide part and a nucleic acid part, each peptide-oligonucleotide conjugate (POC) represents a combination of its parent bio-molecules such as the immanent base-pairing ability of nucleic acids and the multifaceted bioactivity of the structurally and functionally diverse peptides.”
Though this reviewer has only provided one example, this editing suggestion is not likely to be trivial to execute.”
We have amended the paragraph 1 (page 1) according to the Reviewer 1 suggestion. Throughout the text, attempts have been made and changes introduced wherever possible to shorten and comply with the above recommendation. Unfortunately, due to the length of the manuscript’s and strict time constraints, further improvements could only be made by enlisting a professional help.
- In Figure 2, the authors should include the names of conjugates 3-7 in either the figure itself or in the caption.
The names are given in the Figure 2 legend.

Reviewer 2 Report
This review introduces the problems of nucleic acid therapeutics and nucleic acid drug delivery, the types of cell-permeable peptides and the mechanisms of intracellular transport, as well as delivery using non-covalent peptide additives and peptides covalently attached to oligonucleotides. In addition, the methodology for synthesizing covalent complexes of peptides and oligonucleotides for these applications is described in detail with special focus on the methodologies used.
The chemical synthesis of peptide-oligonucleotide conjugates (POCs) is important for the area of nucleic acids therapy, because using of the oligonucleotides conjugated with peptide are one of the strategies for dissolving the problem of nucleic acids delivery. This review is covered over the chemical synthesis of POCs and would result in an extensive review. The manuscript might be suitable for publication after the following points are addressed:
Authors described the chemical synthesis of POCs using the schematic figures and it is difficult to understand them. Author should show the concrete example using synthetic scheme. For example, Figure 4a is difficult to understand.
Minor points:
- In Figure 4b, the starting material should be XH on the solid support.
- In page 15 line 8 from the top, Figure 7 is wrong and should be Figure 8.
- In page 19 line 5 from the top, “t” should be italic in “Nα-Fmoc-O-t-Bu-serine”.
- In page 20 line 10 and line 7 from the bottom, “t” should be italic in “t-Bu”
Author Response
REVIEWER 2
- “Authors described the chemical synthesis of POCs using the schematic figures and it is difficult to understand them. Author should show the concrete example using synthetic scheme. For example, Figure 4a is difficult to understand.”
We have replaced the most general and therefore most ambiguous to read Figures 4 and 9 with redesigned new Figures (pages 13 and 16 respectively). We hope the new Figures are much clearer to understand and aesthetically more appealing. As the old Figure 6 overlapped with the new Figure 4b, it has been deleted, so the total number of Figures dropped down from 20 to 19. The replacement Figure 9 is now numbered Figure 8.
Minor points:
- In Figure 4b, the starting material should be XH on the solid support.
- In page 15 line 8 from the top, Figure 7 is wrong and should be Figure 8.
- In page 19 line 5 from the top, “t” should be italic in “Nα-Fmoc-O-t-Bu-serine”.
- In page 20 line 10 and line 7 from the bottom, “t” should be italic in “t-Bu”
All the minor points have been corrected in the text as well as the typos found.
On behalf of all the authors
Alesya A. Fokina
